# Virtual monochromatic imaging with projection-based material decomposition algorithm for metal artifacts reduction in photon-counting detector computed tomography

Chia-Hao Chang[1,2], Hsiang-Ning Wu[1], Ching-Han Hsu[2]*, Hsin-Hon Lin[3,4,5]*

1 Health Physics Division, Institute of Nuclear Energy Research, Atomic Energy Council, Taoyuan, Taiwan, 2 Department of Biomedical Engineering and Environmental Sciences, National Tsing Hua University, Hsinchu, Taiwan, 3 Department of Medical Imaging and Radiological Sciences, Chang Gung University, Taoyuan, Taiwan, 4 Department of Nuclear Medicine, Keelung Chang Gung Memorial Hospital, Keelung, Taiwan, 5 Institute for Radiological Research, Chang Gung University/Chang Gung Memorial Hospital, Taoyuan, Taiwan

* cghsu@mx.nthu.edu.tw (CHH); hh.lin@mx.nthu.edu.tw (HHL)

**Data Availability Statement:** All relevant data for the simulation and experiments in this work can be

## Abstract

Metal artifacts present a major challenge to computed tomography (CT) because they reduce the image quality in medical diagnosis and treatment. Several metal artifact reduction (MAR) methods have been proposed to address this issue in previous studies. This study aimed to synthesize a virtual monochromatic image for MAR in CT images using projection-based material decomposition (MD) algorithms. We developed a spectral micro-CT prototype system equipped with a photon-counting detector (PCD) and PCD-CT imaging simulator to assess the performances of different MAR methods. Two projection-based MD algorithms were implemented and evaluated for their MAR performances in CT images and compared with conventional sinogram inpainting MAR methods. Different parts of digital 4D-extended cardiac torso (XCAT) phantoms with metal implants were designed to simulate various real scenarios. A homemade metal artifact evaluation (MAE) phantom was used to evaluate the MAR performance in experiments. The simulated results of the XCAT phantom indicated that the projection-based virtual monochromatic CT (VMCT) images provided better image quality than the conventional MAR images without blurring the normal tissues at the position of the metal artifacts. Various quantitative indicators support this conclusion. Additionally, the experimental results of the MAE phantom reveal that projection-based VMCT images can avoid image distortion caused by metal artifacts, unlike conventional MAR methods. In regards to the projection-based VMCT images, the simulated and experimental results demonstrated that using the linear maximum likelihood estimators with an error correction look-up table algorithm yielded better MAR performance compared to that obtained using a polynomial algorithm. Furthermore, projection-based VMCT images can not only reduce metal artifacts effectively but also simultaneously prevents object blurring at the metal artifact position and image distortion of the metal implants. Hence, the CT image

found on Figshare (https://doi.org/10.6084/m9.figshare.22032020).

**Funding:** Atomic Energy Council of Taiwan (Grant No. AIE01030302) National Science and Technology Council of Taiwan (Grant No. 111-2221-E-182 -008 -MY3) Chang Gung Memorial Hospital (Grant No. CMRPD1K0442 and BMRP14). The funders had no role in the study design, data collection, analysis, decision to publish, or preparation of the manuscript.

quality can be further improved to increase the abilities for both preoperative and postoperative assessment of metal implants.

## Introduction

In X-ray computed tomography (CT) imaging, metallic implants, such as dental fillings, bone screws, and hip prostheses, can induce metal artifacts in the reconstructed images [1–3]. Materials containing high atomic numbers (such as metals) cause beam hardening, photon starvation, and scattering, resulting in cupping artifacts, dark streaking between dense objects, and severe dark/bright streak artifacts around the metal implant in the reconstructed CT images [4]. Hence, these mechanisms degrade the CT image quality and thereby the diagnostic performance of the radiologists.

In such a scenario, several metal artifact reduction (MAR) methods have been proposed. For non-removable metal items, the first approach is to adjust the system imaging parameters by increasing the tube voltage, using gantry angulation to exclude the metal inserts from scans of nearby anatomy, and using thin sections to remove the metal artifacts from the image [4]. The second approach is based on image post-processing. One of the most widely known methods is the sinogram inpainting algorithm, in which the metal trace areas are determined by the sinogram of metal items and replaced with suitable correct values [5]. Two major methods are used to fill in the metal trace areas: linear MAR (LMAR) that uses linear interpolation with the surrounding values [6] and normalized MAR (NMAR) [7] that uses the initially corrected images from LMAR as a prior image and normalizes the sinogram to minimize the interpolation errors within the metal trace areas. The prior image quality severely affects the performance of NMAR. To improve the effectiveness of MAR more efficiently, a third approach, which is an iterative-based method, and several other methods have been developed. Various regularization terms, such as total-variation [8], penalized likelihood sinogram smoothing [9], and prior image models [10], have been used to manage the ill-posed conditions in MAR. Compared to the sinogram inpainting methods, the iterative MAR methods are more robust against noise; however, the computation time is a major limitation of the iterative MAR methods. In recent years, a fourth approach, based on deep-learning MAR, has been proposed. These methods are mostly based on convolutional neural networks (CNN) and conditional generative adversarial networks that are applied in the image or sinogram domain. In the image domain approach, a CNN is used to reduce the residual errors of the conventional MAR result image and reference image to obtain better results [11]. In the sinogram domain approach, a CNN is used to estimate the missing data in the metal trace areas of the sinogram [12]. Although deep-learning methods have shown remarkable MAR results, they require a large amount of data for neural network training.

Spectral CT is an emerging technology with the potential to dramatically change the field of clinical CT. Various approaches have been proposed to achieve dual-energy spectral CT, such as dual-source, fast tube voltage switching, dual-layer scintillator, and dual-filters techniques [13–15]. Recently, spectral CT equipped with a photon-counting detector (PCD) has been introduced into pre-clinical animal and clinical CT imaging fields [16–18]. PCD-based multi-energy spectral CT imaging is an innovative technology [19]. Compared with the energy-integrating detector of the conventional dual-energy spectral CT, PCD-CT has many advantages, such as reducing beam-hardening effects, improving material decomposition, avoiding spectral overlapping, and multi-energy imaging capability [20].

Several spectral CT applications have been proposed. Virtual monochromatic imaging (VMI) is an application of spectral CT that has the potential to reduce beam hardening or metal artifacts and can provide more quantitative attenuation information [2, 21, 22]. VMI can be synthesized using projection-domain [23, 24] or image-domain methods [25, 26] based on material decomposition (MD). The advantage of image-based VMI is that it is easy to implement, but the reconstructed CT image is severely affected by image artifacts [27]. Compared to image-based MD, the projection-based MD method offers more accurate image quantification of contrast agents and fewer artifacts caused by beam hardening [26]. However, an additional calibration process is required [28]. For VMI in clinical spectral CT, image-based MD is usually used because it is easy to implement [29]. Despite image-based MD being computationally faster, projection-based MD provides better MD results [26]. Furthermore, projection-based MD is the more precise model that directly decomposes projection data rather than after the procedure of reconstruction [30]. Hence, projection-based MD is more suitable to generate VMI with the ability of metal artifact reduction in the CT images.

To the best of our knowledge, limited research has been conducted on the effectiveness of projection-based MD in reducing metal artifacts in PCD-CT images [28, 31]. In this study, we developed two different projection-based MD algorithms to generate virtual monochromatic CT (VMCT) images and compared them with the conventional MAR methods (LMAR and NMAR) using various quantitative indicators. To evaluate the MAR performance of these methods, we developed a PCD-CT imaging simulator with a 4D-extended cardiac torso (XCAT) digital phantom [32] and built a spectral micro-CT prototype system equipped with a PCD to obtain simulation and experimental data. We also assessed the influence of the CT metal artifacts on the metal implants and their surrounding materials using the abovementioned MAR methods.

## Materials and methods

### Material decomposition model

In the diagnostic energy range, it is assumed that the linear attenuation coefficient can be approximated well by a linear combination of two basis functions [33]:

$$\mu(r, E) = a_{m1}(r)f_{m1}(E) + a_{m2}(r)f_{m2}(E), \quad (1)$$

where $\mu(r, E)$ is the linear attenuation coefficient of the object at position $r$ and for energy $E$; $f_{m1}(E)$ and $f_{m2}(E)$ are the basis functions for materials $m1$ and $m2$, respectively; and $a_{m1}(r)$ and $a_{m2}(r)$ are the set coefficients of the basis function for materials $m1$ and $m2$, respectively.

The measured photon number $N_k$ at a PCD pixel can be expressed as:

$$N_k = \int S(E)e^{-\int \mu(r,E)ds}D_k(E)dE, \quad (2)$$

where the subscript $k$ indicates the measurement with the $k$th energy bin; $S(E)$ is the incident photon number of the spectrum at photon energy $E$; $s$ is the path length in which the X-ray penetrates through the object; and $D_k(E)$ is the energy response function of the PCD with the $k$th energy bin at photon energy $E$.

The line integral of Eq (2) can be expressed as:

$$\int \mu(r, E)ds = A_{m1}f_{m1}(E) + A_{m2}f_{m2}(E), \quad (3)$$

where $A_{mi} = \int a_{mi}(r)ds$, i = 1, 2. The line integrals $A_{m1}$ and $A_{m2}$ can be considered to be the effective thicknesses along the ray directions for materials $m1$ and $m2$, respectively.

Substituting the expression for the line integral (Eq (3)) in Eq (2), the measured photon number with the $k$th energy bin can be expressed as:

$$N_k(\mathbf{A}) = \int S(E)e^{-A_{m1}f_{m1}(E)-A_{m2}f_{m2}(E)}D_k(E)dE. \tag{4}$$

The logarithm projection of the measurements can be expressed as:

$$P_k = -ln\left(\frac{N_k}{N_{0k}}\right) = -ln\left(\frac{\int S(E)e^{-A_{m1}f_{m1}(E)-A_{m2}f_{m2}(E)}D_k(E)dE}{\int S(E)D_k(E)dE}\right), \tag{5}$$

where $N_{0k}$ is the measured photon number in the blank scan. The logarithm projections $P_k = -ln(N_k/N_{0k})$ can be approximately linearized with the effective thicknesses $A_{m1}$ and $A_{m2}$ [34]. The purpose of the projection-based MD method is to solve $A_{m1}$ and $A_{m2}$ in Eq (5).

## Projection-based MD algorithms and VMI

There are two major processes for obtaining VMI projections for MAR using projection-based MD algorithms. The first step is the MD calibration process, and the second step is the MD testing process. In the MD calibration process, the material characteristics and thickness of the calibration phantom are determined. Calibration data were acquired using an MD calibration phantom with suitable imaging parameters. Subsequently, the characteristics of the calibration phantom and calibration data were used as the input of the MD calibration procedure to generate the system model for the MD. In the MD testing process, the input is the testing object, which is used to obtain the VMI projections. Various energy bin projections of the testing object were acquired using the same imaging parameters as the MD calibration process. The varied energy bin projections of the testing object and the system model of the MD calibration process were applied to the projection-based MD algorithm to generate different basis material projections. Subsequently, the basis material projections were applied to the VMI algorithm with VMI projections created using specific photon energies. The flowchart of the proposed method is shown in Fig 1. The procedure and algorithms (e.g., the MD calibration procedure, basis MD, and VMI algorithm) for MAR are introduced in the following section.

Two projection-based MD algorithms are introduced in this study to solve $A_{m1}$ and $A_{m2}$ because the physical model used in Eq (5) is complex and is related to the properties of the X-ray tube, PCD, and objects. In the first projection-based MD algorithm (hereinafter called VMI-Poly), to simplify the model with two energy bins, Eq (5) can be rewritten in a

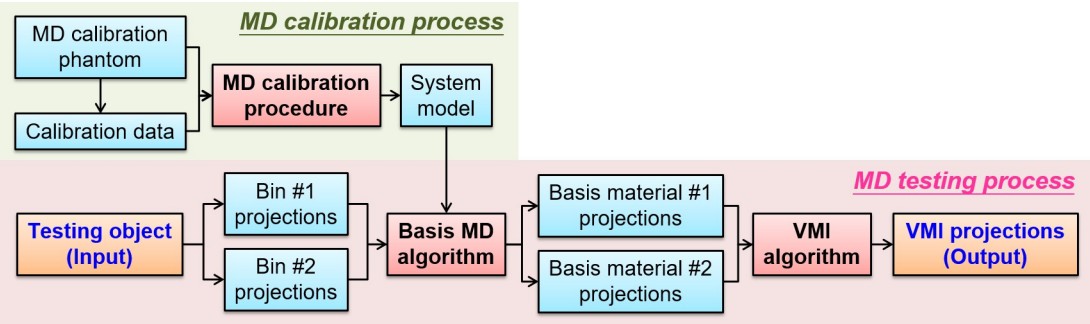

**Fig 1. Scheme of VMI synthesis for MAR using projection-based MD algorithms.**

polynomial approximation by Taylor expansion as follows [35]:

$$P_1 = c_{11} + c_{12}A_{m1} + c_{13}A_{m2} + c_{14}A_{m1}^2 + c_{15}A_{m1}A_{m2} + c_{16}A_{m2}^2, \tag{6}$$

$$P_2 = c_{21} + c_{22}A_{m1} + c_{23}A_{m2} + c_{24}A_{m1}^2 + c_{25}A_{m1}A_{m2} + c_{26}A_{m2}^2, \tag{7}$$

where $c_{11}, \ldots, c_{16}$ and $c_{21}, \ldots, c_{26}$ are undetermined coefficients and $P_1$ and $P_2$ are the acquired projections with energy bin numbers of 1 and 2, respectively. Eqs (6) and (7) can be reformulated in the reverse form to obtain the effective thicknesses $A_{m1}$ and $A_{m2}$ [36]

$$A_{m1} = d_{11} + d_{12}P_1 + d_{13}P_2 + d_{14}P_1^2 + d_{15}P_1P_2 + d_{16}P_2^2, \tag{8}$$

$$A_{m2} = d_{21} + d_{22}P_1 + d_{23}P_2 + d_{24}P_1^2 + d_{25}P_1P_2 + d_{26}P_2^2, \tag{9}$$

where $d_{11}, \ldots, d_{16}$ and $d_{21}, \ldots, d_{26}$ are undetermined coefficients. The model is completely constructed when the coefficients are determined. Moreover, the coefficients can be determined using sufficient calibration data obtained from the MD calibration phantom composed of the known thicknesses $m1$ and $m2$ under two different X-ray energy bins. Hence, the effective thickness of the basis materials $A_{m1}$ and $A_{m2}$ in the object can be calculated from the two energy bin projections, $P_1$ and $P_2$. This process is called projection-based MD, and the two energy bin projections can be decomposed into two basis images using the polynomial approximation method.

The second projection-based MD algorithm (hereinafter called VMI-Atable) uses linear maximum likelihood estimators (MLE) with a look-up table (LUT) error correction, which is the so-called Atable approximation method. For more details, refer to [34]. This method has a fast computational advantage over the iterative-based method. The MLE requires a probabilistic model consisting of the measured projections **P** (consisting of $P_1$ and $P_2$) and the effective thickness of the basis materials **A** (consisting of $A_{m1}$ and $A_{m2}$) with a multivariate normal distribution [37]. Therefore, the linear model with noise is defined as:

$$\mathbf{P}(\mathbf{A}) = \mathbf{MA} + \mathbf{w}, \tag{10}$$

where **w** is a zero-mean multivariate normal random variable whose covariance depends on **A** and **M** is the average or effective linear attenuation coefficient matrix.

Furthermore, the initial estimated effective thickness $\mathbf{A}_{MLE}$ of the basis material can be obtained using the linear MLE model, as follows:

$$\mathbf{A}_{MLE} = (\mathbf{M}^T \mathbf{R}_{P|A}^{-1} \mathbf{M})^{-1} \mathbf{M}^T \mathbf{R}_{P|A}^{-1} \mathbf{P}, \tag{11}$$

where $\mathbf{R}_{P|A}$ is the covariance of the projections **P**; Matrix **M** is determined from the MD calibration process $\mathbf{P}_{calib} = \mathbf{MA}_{calib}$, where $\mathbf{P}_{calib}$ and $\mathbf{A}_{calib}$ are the measured projections and the known thicknesses of the calibration phantom $m_1$ and $m_2$, respectively.

The MD calibration data can be used to compute the error correction LUT. The initial estimation results of the MLE were expected to produce errors owing to the nonlinear relationship between the measured projections **P** and thickness **A** of the basis materials. During the MD calibration process, the actual thicknesses $\mathbf{A}_{calib}$ of the calibration phantom are determined, and we can obtain the initial estimated effective thicknesses $\mathbf{A}_{MLE,calib}$ of the calibration phantom using Eq (11). By subtracting the actual thickness values $\mathbf{A}_{calib}$ obtained from the estimated linear MLE results $\mathbf{A}_{MLE,calib}$ from the known calibration phantom thicknesses, we can compute the error correction values $\delta\mathbf{A}$. Hence, we can use a series of error correction values

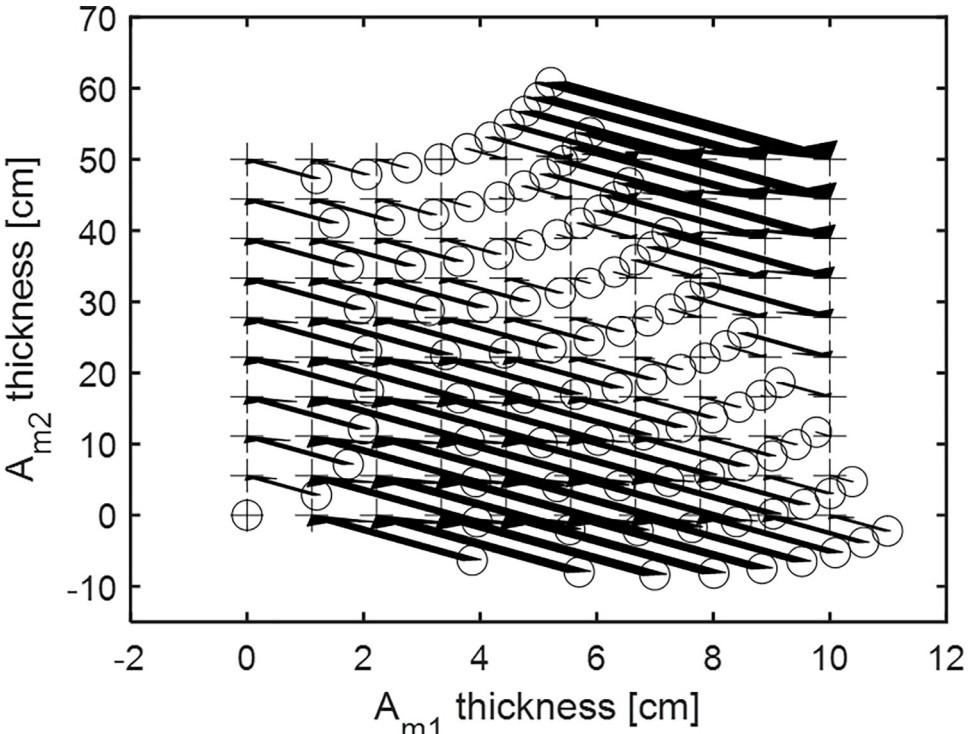

**Fig 2. The error correction LUT is calculated from the MD calibration process.** The circles represent $\mathbf{A}_{MLE,calib}$, and the crosses represent $\mathbf{A}_{calib}$. The arrows represent the error correction values $\delta\mathbf{A}$ for various material thicknesses.

$\delta\mathbf{A}$ obtained with various thickness values $\mathbf{A}_{calib}$ to construct the error correction LUT.

$$\delta\mathbf{A} = \mathbf{A}_{calib} - \mathbf{A}_{MLE,calib} \tag{12}$$

For example, the error correction LUT obtained from the MD calibration process is shown in Fig 2. The circles represent the initial estimated effective thickness $\mathbf{A}_{MLE,calib}$ of the calibration phantom, and the crosses represent the actual thickness of the calibration phantom $\mathbf{A}_{calib}$. Two materials, $m_1$ and $m_2$ are used for the calibration phantom. The arrows denote the error correction values $\delta\mathbf{A}$.

The effective thicknesses of the basis materials of an object can be estimated using linear MLE with error correction LUT, as follows:

$$\mathbf{A} = \mathbf{A}_{MLE} + \delta\mathbf{A}, \tag{13}$$

where $\mathbf{A}$ comprises $A_{m1}$ and $A_{m2}$, $\mathbf{A}_{MLE}$ is the initial estimated thickness of the basis materials computed from the measured projections $\mathbf{P}$ of the objects with two energy bins using Eq (11), and $\delta\mathbf{A}$ is the error correction value for $\mathbf{A}_{MLE}$ calculated from the MD calibration process.

The 2D VMI can be synthesized using the effective thicknesses $A_{m1}$ and $A_{m2}$ of the basis materials and linear attenuation coefficients at specific X-ray photon energies as follows [23]:

$$\mathbf{P}(E_{VMI}) = \mu_{m1}(E_{VMI}) \cdot A_{m1} + \mu_{m2}(E_{VMI}) \cdot A_{m2}, \tag{14}$$

where $E_{VMI}$ denotes the specific X-ray photon energy and $\mu_{m1}$ and $\mu_{m2}$ are the linear attenuation coefficients of materials $m1$ and $m2$ at energy $E_{VMI}$, respectively. The linear attenuation coefficients $\mu_{m1}$ and $\mu_{m2}$ of the specific energy and material can be found in the photon cross-

section database of NIST-XCOM [38]. The VMI can then be used as input to the CT image reconstruction algorithm to obtain the VMCT images.

## Spectral micro-CT prototype system using a PCD

We developed a spectral micro-CT prototype system equipped with a PCD (XC-Thor, Direct Conversion, Sweden) and a microfocus X-ray tube (L12161-07, Hamamatsu, Japan). The X-ray tube provides three focal spot modes (7, 20, and 50 μm) and can operate with a voltage of up to 150 kV and a power of up to 75 W. The PCD equips with a 0.75-mm thick cadmium telluride (CdTe) sensor and provides two energy bins by setting two energy thresholds (low and high energy thresholds). The anti-coincidence mode of the PCD was used to eliminate the charge-sharing effect of the PCD in all experiments. The active detection region of PCD is $51.2 \times 102.4$ mm$^2$ with a dimension of $512 \times 1024$ pixels (the pixel size is 100 μm$^2$). For the tomographic scan, a direct-drive servo motor (SGMCS-02B3C11, Yaskawa, Japan) was used to move the subject through all rotation angles.

A photograph of the prototype PCD micro-CT system is shown in Fig 3. In the system configurations, the source-to-axis distance (SAD) and source-to-image-receptor distance (SID) were 365 and 770 mm, respectively, with a magnification of approximately 2.11. The experimental projections were obtained at a tube voltage of 140 kV, a tube current of 155 μA, and an exposure time of 400 ms. For the scanning protocol in the experiment, we obtained 360 projections within 360°. The low-energy and high-energy threshold was set at 30 keV and 80 keV, resulting in two energy bins of [31 80] and [81 140] keV. The images were then reconstructed using the Feldkamp–Davis–Kress (FDK) algorithm [39] in a matrix of size $512 \times 512 \times 512$ pixels with a voxel size of $0.1 \times 0.1 \times 0.1$ mm$^3$.

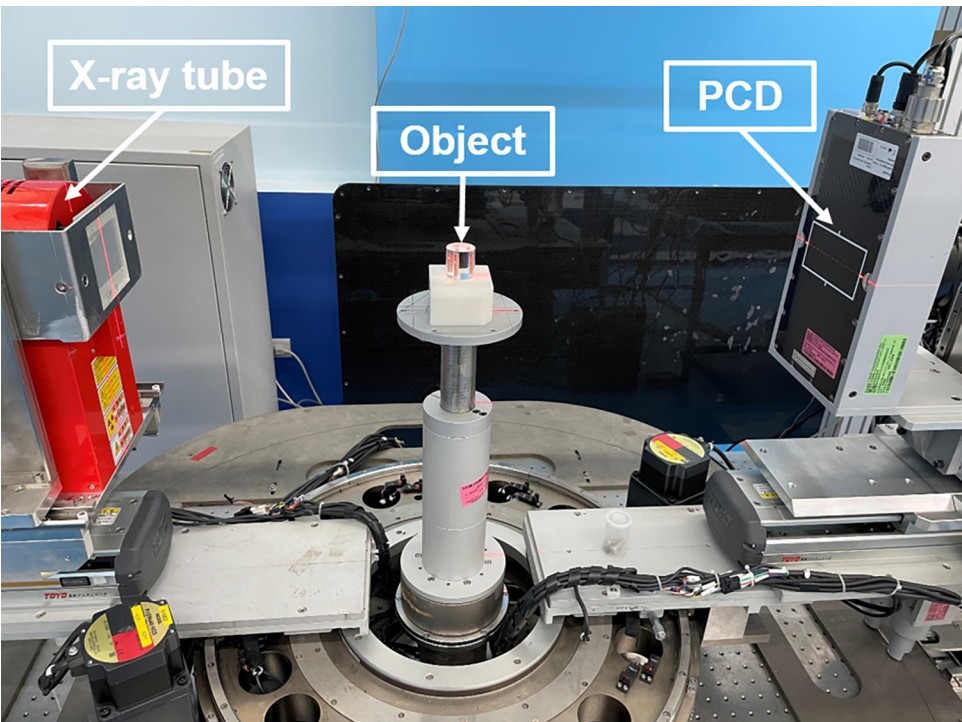

**Fig 3. Image of the spectral micro-CT prototype system with PCD.**

### The simulation setup for PCD-CT imaging

To evaluate the performance of the various MAR methods, we developed a program to simulate the two-dimensional (2D) spectral X-ray projections in a specific energy range at different angles from the three-dimensional (3D) digital objects [40, 41]. A numerical method was used to simulate the PCD-CT imaging, which is based on the ray-tracing algorithm for calculating line integrals [42]. The ray-tracing algorithm kernel of the C++ program from the ASTRA-toolbox package was used to simulate the X-ray forward projections while being linked to MATLAB using the MATLAB C++ MEX function [43]. To improve the simulation efficiency, the focal spot of the X-ray tube was assumed to be an ideal point source, and only the primary photons (simple integration of the path) were simulated. Moreover, the PCD exhibits perfect energy resolution and detection efficiency. The input parameters of this program consist of the X-ray tube energy spectrum, object materials and geometry, PCD properties, imaging acquisition parameters, and geometry of the PCD-CT imaging system. The outputs were 2D spectral X-ray projections in a specific energy range with various rotation angles. A polyenergetic X-ray beam was generated using the SPEKTR 3.0 toolbox to obtain realistic spectral projections [44].

The energy spectrum used was based on the tungsten anode spectral model using interpolating polynomials (TASMIP) model with 1.6-mm thick aluminum (Al) inherent filtration [45]. The voltage of the X-ray tube used for simulating the spectral projections was 140 kV, with an additional 2-mm thick Al filtration, as shown in Fig 4.

The detailed parameters of the PCD-CT imaging simulations are listed in Table 1. Two fan-beam CT geometries were used to validate the performance of the various MAR methods for both dental and conventional PCD-CT imaging systems. Each PCD pixel was expected to receive $10^7$ photons in the case of a blank scan, and each measurement followed the Poisson distribution [46]. For spectral X-ray projection generation, an energy range of 1–150 keV with a step size of 1 keV was used to simulate the interaction between the photons and materials. The settings of the low- and high-energy thresholds for the PCD were 20 and 90 keV, respectively.

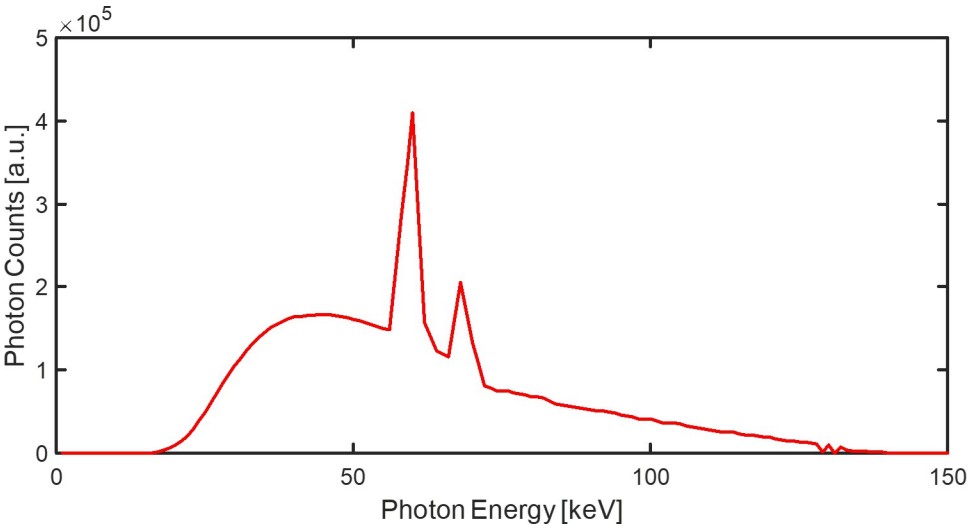

**Fig 4. TASMIP X-ray spectrum at a tube voltage of 140 kV with 2-mm thick Al filtration.**

**Table 1. The parameters for data acquisition and image reconstruction for the PCD-CT imaging simulations.**

| Parameters | Dental PCD-CT geometry | Conventional PCD-CT geometry |
|---|---|---|
| Source to axis distance (SAD) | 400 mm | 600 mm |
| Axis to image receptor distance (AID) | 200 mm | 400 mm |
| Magnification | 1.5 | 1.67 |
| Number of views | 1080 views | |
| Image receptor pixel size | $0.5 \times 0.5$ mm$^2$ | $1 \times 1$ mm$^2$ |
| Image receptor number | 1024 pixels | |
| Reconstructed algorithm | Filtered back-projection (FBP) | |
| Reconstructed image size | $256 \times 256$ mm$^2$ | $512 \times 512$ mm$^2$ |
| Reconstructed matrix size | $512 \times 512$ pixels | |
| Reconstructed pixel size | $0.5 \times 0.5$ mm$^2$ | $1 \times 1$ mm$^2$ |

## Digital XCAT phantoms

To validate the performances of the MAR methods, regions of the human head, abdomen, and hip were generated using the XCAT phantom model [32]. Three different types of titanium (Ti) metal implants were created as binary images by setting the shape and position as similar to the actual metal implants that would be present in the human body. The three metal images were then inserted into the original XCAT images of the different regions. Linear attenuation coefficients of materials in the XCAT phantom were referred from the NIST-XCOM database [38].

The detailed parameters of the PCD-CT imaging simulations and XCAT phantoms are listed in Tables 1 and 2, respectively. The head XCAT phantom was used for the dental PCD-CT imaging simulation, and the abdominal and hip XCAT were used for the conventional PCD-CT imaging simulation. Fig 5 shows the XCAT phantoms and the reconstructed CT images. The XCAT images were presented using linear attenuation coefficients at a photon energy of 120 keV. The metal implant positions are indicated by the yellow regions. The reconstructed CT images of the XCAT phantoms were simulated using a 140 kV polychromatic TASMIP energy spectrum. Reference images without metal artifacts were simulated using 100 keV monochromatic energy, which could be used as a gold standard.

## Phantoms for acquiring MD calibration data and evaluating metal artifacts in the experiments

To obtain the system model of the MD calibration process (as shown in Fig 1), the MD calibration phantoms were used to acquire the calibration data in the simulation and experiment cases. In the simulated case, the calibration phantom consisted of polymethyl methacrylate (PMMA) and Al with different known thicknesses. The thicknesses of the PMMA and Al ranged from 0 to 50 cm and 0 to 10 cm, respectively. The different thicknesses were generated in 10 steps with equal increments for each material. In the experimental case, the calibration phantom consisted of PMMA and copper (Cu) of thicknesses in the range 0–29.75 mm and 0–10 mm, respectively. The area of these materials was approximately 100 mm × 100 mm. The

**Table 2. The parameters for various regions of the XCAT phantom.**

| Parameters | Head | Abdomen | Hip |
|---|---|---|---|
| Voxel size | 0.25 mm$^3$ | 0.4 mm$^3$ | 0.5 mm$^3$ |
| Phantom dimension | $256\times256\times0.25$ mm$^3$ | $409.6\times409.6\times0.4$ mm$^3$ | $512\times512\times0.5$ mm$^3$ |

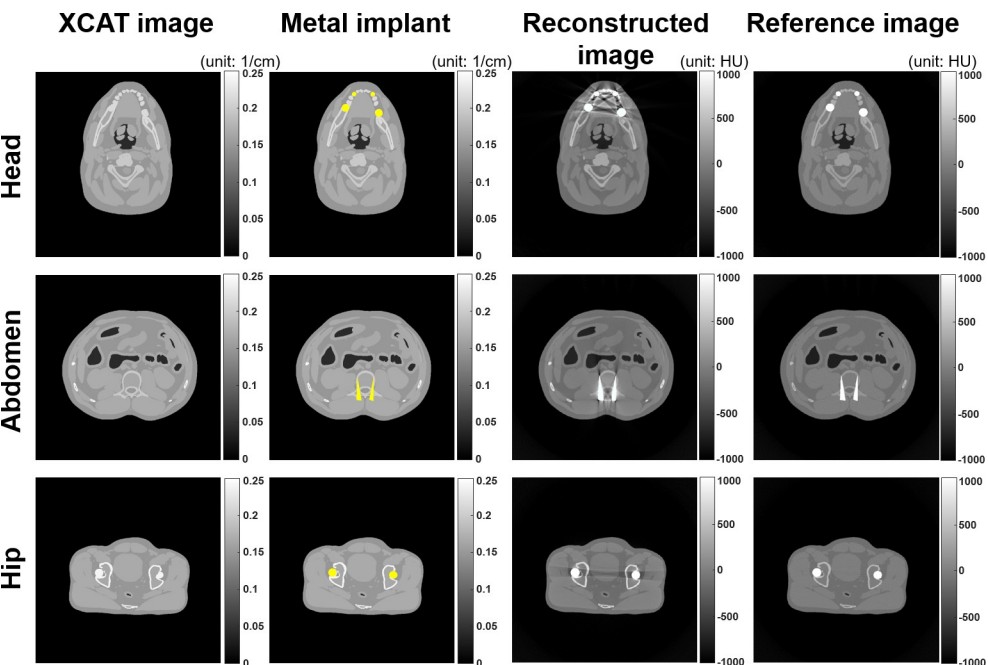

**Fig 5. Representative images of the XCAT phantoms.** Each row presents various parts of the human body. Each column presents the original XCAT image, metal implant positions (indicated by the yellow regions), reconstructed CT image with metal artifacts, and reference image without metal artifacts. The display unit for the XCAT image and metal implant is cm$^{-1}$. For reconstructed images and reference images, the display unit is the CT value in HU.

calibration phantom and thickness combinations used in the experiments are shown in Fig 6. Nine and six thickness combinations were used for the Cu and PMMA, respectively. To evaluate the performances of the MAR methods in the experiments, we designed a metal artifact evaluation (MAE) phantom. The MAE phantom consisted of three Cu rods and was covered with a PMMA cylinder, as shown in Fig 7. Copper was chosen because it can induce severe metal artifacts in the CT images. The three Cu rods were placed symmetrically in the center of the PMMA. The diameter and height of the Cu rods were 5 mm and 24 mm, respectively. The diameter and height of the PMMA cylinders were 30 mm and 30 mm, respectively.

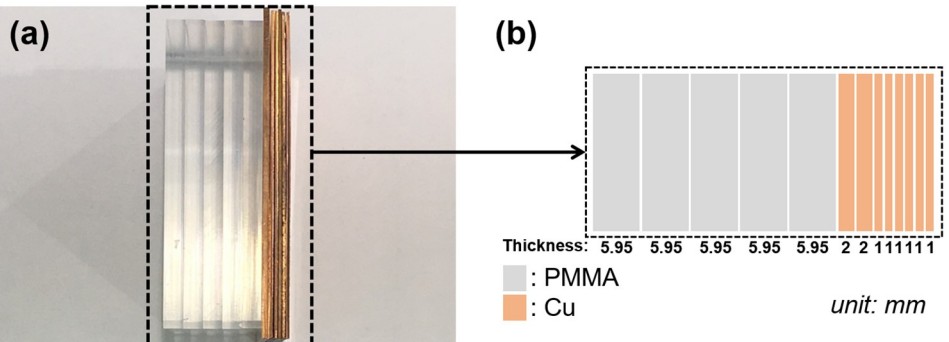

**Fig 6. MD calibration phantom in the experiments.** (a) Front view of the calibration phantom. (b) Schematic of the thickness combination of the PMMA and copper.

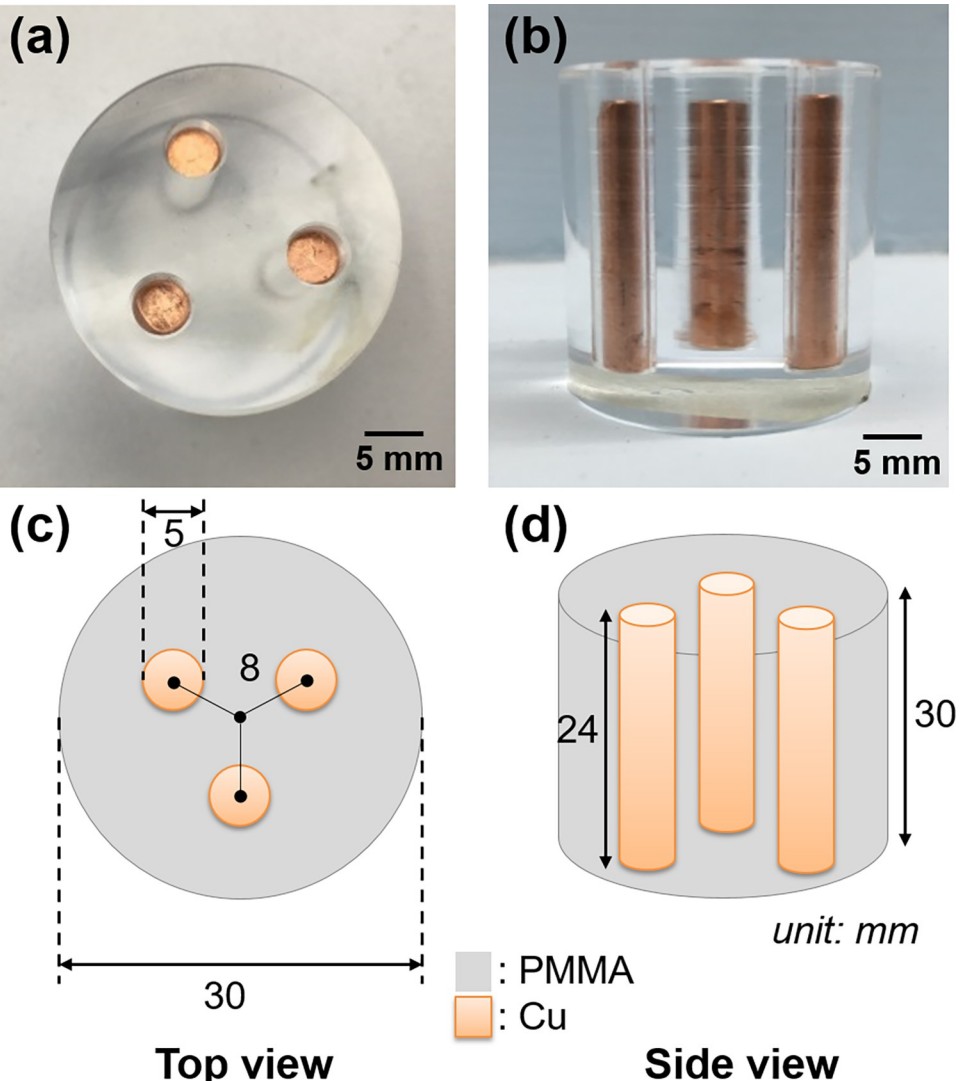

**Fig 7. Metal artifacts evaluation (MAE) phantom in the experiments.** (a) Top view of the phantom. (b) Side view of the phantom. (c) Schematic of the cylinder diameters and positions in the phantom. (d) Schematic of the cylinder heights in the phantom.

## Image quality evaluation

Three quantitative indicators were used to evaluate the image quality and performance of the different MAR methods in the simulations, including peak signal-to-noise ratio (PSNR), normalized root-mean-square error (NRMSE), and structural similarity index measure (SSIM). PSNR and NRMSE denote the difference in the gray values (HU values of CT images) between the two images, and SSIM is a well-known quality metric used for measuring the similarity between two images. The PSNR is defined as follows:

$$\text{PSNR}(f, g) = 10 \cdot \log_{10}\left(\frac{max(f)^2}{MSE(f, g)}\right), \tag{15}$$

$$MSE(f, g) = \frac{1}{M \times N}\sum\nolimits_{i=1}^{M}\sum\nolimits_{j=1}^{N}(f_{ij} - g_{ij})^2, \tag{16}$$

where *f* is the reference image and *g* is the corrected image. The image size of *f* and *g* is *M*×*N* and the pixel positions are *i* and *j*, respectively.

The NRMSE is defined as:

$$NRMSE(f,g) = \sqrt{\frac{\sum_{i=1}^{M} \sum_{j=1}^{N} (f_{ij} - g_{ij})^2}{\sum_{i=1}^{M} \sum_{j=1}^{N} (g_{ij})^2}}. \tag{17}$$

The SSIM is defined as follows:

$$SSIM(f,g) = \frac{(2\mu_f \mu_g + C_1)(2\sigma_{fg} + C_2)}{(\mu_f^2 + \mu_g^2 + C_1)(\sigma_f^2 + \sigma_g^2 + C_2)}, \tag{18}$$

where $\mu_f$, $\sigma_f$, $\mu_g$, and $\sigma_g$ denote the average gray values and standard deviations of the images *f* and *g*; $\sigma_{fg}$ is the covariance between the images *f* and *g*; and $C_1$ and $C_2$ indicate the coefficients for the SSIM calculation and range from 0 to 1. The quantitative indicators mentioned above were calculated between the reference image and corrected images using the MAR methods.

In the experiments, the circularity was measured on the rods of MAE phantom images to evaluate the influence of the different MAR methods on image distortions in the metal artifact regions. We calculated the circularity of the metal rods in the reconstructed image of the MAE phantom by using the different MAR methods. Circularity is defined as follows:

$$Circularity = \frac{4\pi \times Area}{Perimeter^2}, \tag{19}$$

where *Area* and *Perimeter* denote the area and perimeter of the rod, respectively. The value of circularity lies between 0 and 1, and the value has a larger deviation from 1 for greater non-circular shapes.

## Results

### Simulation results of digital XCAT phantoms for the MAR methods and projection-based VMI algorithms

The results of the two conventional MAR methods, LMAR and NMAR, and the two projection-based VMI algorithms, VMI-Poly and VMI-Atable, for MAR with the XCAT phantom are presented in Fig 8. Each row of Fig 8 corresponds to the head, abdomen, and hip region. The first column represents the reference image for each case, which is obtained by using monochromatic energy of 100 keV without metal artifacts. The second column represents the uncorrected images obtained using a polychromatic TASMIP energy spectrum of 140 kV. The third to sixth columns represent the corrected images obtained using the conventional MAR methods (LMAR and NMAR) and projection-based VMI algorithms (VMI-poly and VMI-Atable). The enlarged images of the region of interest (ROI) in Fig 8 are displayed in Fig 9 for a detailed comparison.

Fig 9 shows the corrected images of the different MAR methods that can reduce metal artifacts. However, the projection-based VMIs provided better performance for MAR compared to the conventional MAR methods, as indicated by the red and orange arrows for representing the different scenarios. The blurred normal tissues at the metal artifact positions of the corrected images with the LMAR and NMAR methods can be observed because of the sinogram inpainting process with numerical interpolation of the conventional MAR methods (red arrows). Nevertheless, the normal tissues around the metal implants can be clearly observed using projection-based VMI algorithms (orange arrows). Furthermore, it can be noticed that

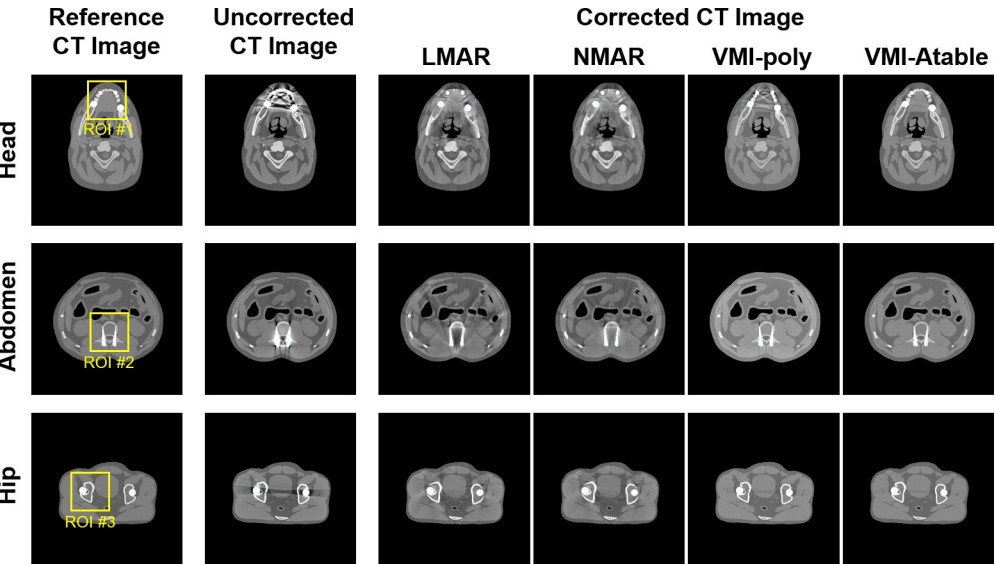

**Fig 8. Simulated results using the XCAT phantom for evaluating the different MAR methods.** Each row corresponds to a different part of the human body. The columns represent the reference CT image; uncorrected CT image with metal artifacts; and corrected CT images obtained using LMAR, NMAR, and the projection-based VMI algorithms (VMI-poly and VMI-Atable) at 100 keV. Window width and level of the image display are 1,000 HU and 0 HU, respectively.

the VMI-Atable method reduces metal artifacts more effectively than the VMI-poly method in certain specific cases (e.g. the multiple metal inserts within a small region), as indicated by the green circles in Fig 9.

The results of the different indicators (PSNR, NRMSE, and SSIM) used in the quantitative analysis of the MAR methods are summarized in Table 3. Entire images and three ROIs (ROI

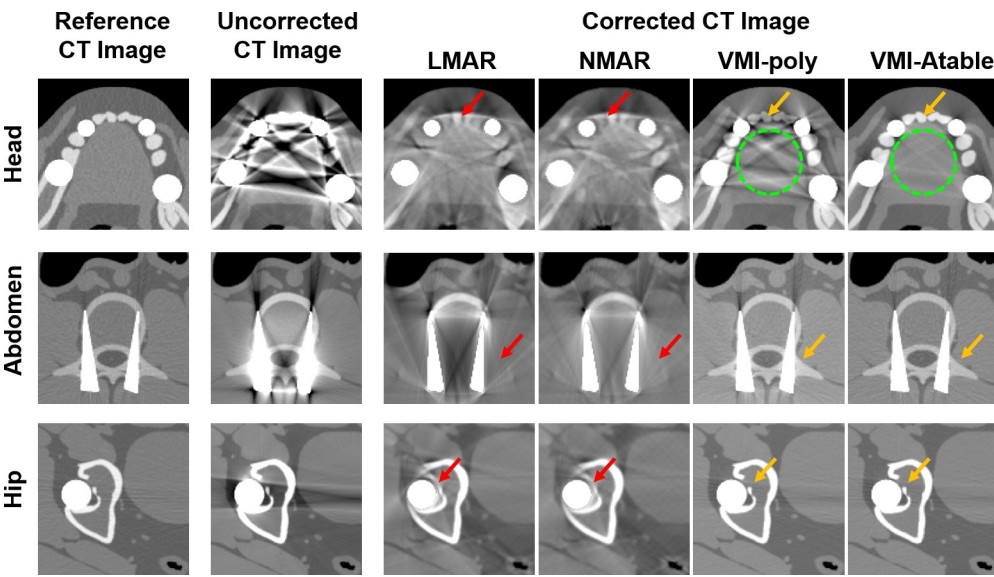

**Fig 9. Enlarged images of the three ROIs in Fig 8.** Each row corresponds to a different part of the human body. The columns represent the reference CT image; uncorrected CT image with metal artifacts; and corrected CT images using LMAR, NMAR and projection-based VMI algorithms (VMI-poly and VMI-Atable) at 100 keV. Window width and level of the image display are 1,000 HU and 0 HU, respectively.

**Table 3. Quantitative data of the MAR methods with XCAT phantom.**

| | | | LMAR | NMAR | VMI-poly | VMI-Atable |
|---|---|---|---|---|---|---|
| Head | whole image | PSNR | 15.2683 | 15.2640 | 18.8480 | 24.8044 |
| | | NRMSE | 0.0220 | 0.0220 | 0.0146 | 0.0073 |
| | | SSIM | 0.7518 | 0.7410 | 0.8498 | 0.9061 |
| | ROI #1 | PSNR | 7.3404 | 7.6529 | 8.5231 | 15.6002 |
| | | NRMSE | 0.0548 | 0.0528 | 0.0478 | 0.0212 |
| | | SSIM | 0.6591 | 0.6739 | 0.7524 | 0.9243 |
| Abdomen | whole image | PSNR | 14.6087 | 16.1153 | 19.0196 | 19.2158 |
| | | NRMSE | 0.0237 | 0.0199 | 0.0143 | 0.0140 |
| | | SSIM | 0.7793 | 0.7831 | 0.8840 | 0.8980 |
| | ROI #2 | PSNR | 5.6701 | 7.4020 | 11.5224 | 12.7297 |
| | | NRMSE | 0.0739 | 0.0606 | 0.0377 | 0.0328 |
| | | SSIM | 0.5805 | 0.7101 | 0.8870 | 0.9068 |
| Hip | whole image | PSNR | 18.0742 | 17.9334 | 18.2371 | 20.8834 |
| | | NRMSE | 0.0159 | 0.0162 | 0.0156 | 0.0115 |
| | | SSIM | 0.6361 | 0.6260 | 0.7469 | 0.8344 |
| | ROI #3 | PSNR | 10.7807 | 11.1213 | 12.4013 | 14.2565 |
| | | NRMSE | 0.0419 | 0.0403 | 0.0348 | 0.0281 |
| | | SSIM | 0.6861 | 0.6928 | 0.7912 | 0.8156 |

#1, ROI #2, and ROI #3) in the reference images were used for the comparison (as shown in Fig 8). For the different body parts, scenarios, the projection-based VMI algorithms (VMI-poly and VMI-Atable) exhibit better performance in regards to the PSNR indicator (with a higher PSNR value) than the conventional MAR methods (LMAR and NMAR), regardless of the entire image or specific ROI. Furthermore, in regards to NRMSE indicator, the projection-based VMI algorithms exhibit better performance (with a lower NRMSE value) than the conventional MAR methods, regardless of the entire image or specific ROI. With respect to the SSIM indicator, the projection-based VMI algorithms exhibit better performance (the value is close to 1) than the conventional MAR methods, regardless of the entire image or specific ROI. The three quantitative indicators demonstrate that the projection-based VMI algorithms (VMI-poly and VMI-Atable) perform better than the conventional MAR methods (LMAR and NMAR) with respect to MAR in CT images. Furthermore, the analysis results also indicate that VMI-Atable performs better than VMI-poly in regards to MAR in CT images.

## Experimental results of the MAE phantom for the MAR methods and projection-based VMI algorithms

The experimental results obtained using the MAE phantom for evaluating the MAR methods are shown in Fig 10. The experimental PCD-CT imaging and reconstruction parameters are provided in the Materials and Methods section. The central slice (slice number: 256) of the CT images was used for the comparison analysis presented in this section. Fig 10 reveals view-aliasing artifacts due to insufficient projections within one rotation, but further collect more projections can minimize the effect. Fig 10(A) shows severe streak artifacts between the two Cu rods, which was a result of the photon beam-hardening effect. Fig 10(B)–10(E) show varying degrees of reduction for the metal artifacts with the different MAR methods. All MAR methods compared in this study demonstrated effectiveness in reducing metal artifacts. To further assess the differences between the MAR methods considered in this study, the line profiles in the CT images for the MAR methods were compared, as shown in Fig 11. Fig 11(A)

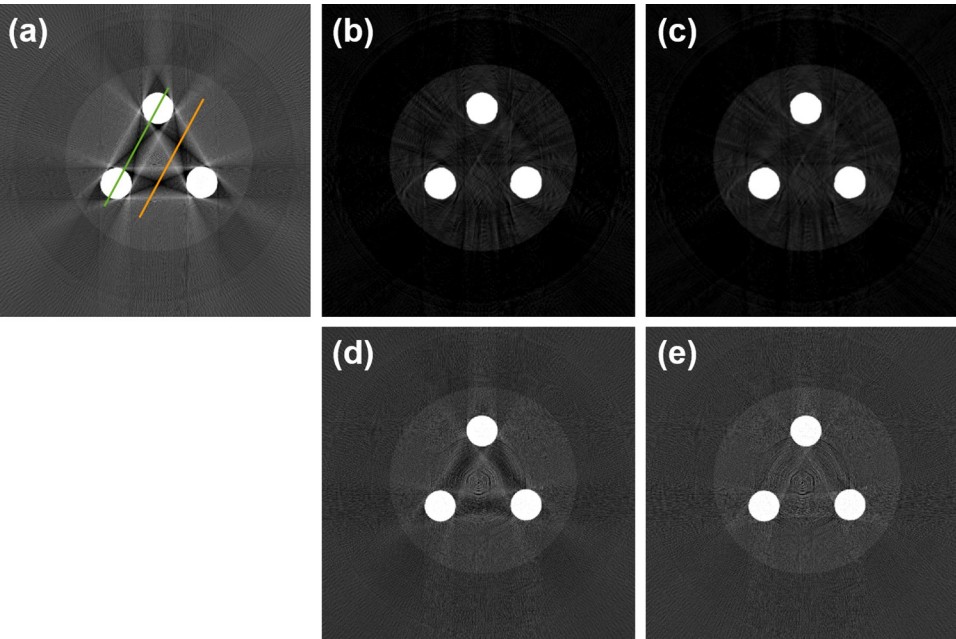

**Fig 10. Experimental results using MAE phantom for evaluating the MAR methods.** The green and orange lines denote different line-profile positions for the analysis. (a) The uncorrected CT image with 140 kV polychromatic X-ray. (b) The corrected CT images obtained using the LMAR method. (c) The corrected CT images obtained using the NMAR method. (d) The projection-based VMCT image obtained using a polynomial method (VMI-poly) at 100 keV. (e) The projection-based VMCT image obtained using the MLE and LUT method (VMI-Atable) at 100 keV. Window width and level of the image display are 11,000 HU and 4,500 HU, respectively.

shows severe cupping artifacts (red arrows) and bright/dark streaks (purple arrows) induced by the high-density materials in the uncorrected CT image. Fig 11(B) and 11(C) show near perfect line profiles of the CT image corrected with the LMAR and NMAR methods, respectively. However, it is difficult to distinguish the object from the surrounding materials when the object is located in the position of the bright/dark streaks (purple arrows in Fig 11(B) and 11(C)) owing to the sinogram inpainting with the interpolating process of LMAR and NMAR (as indicated by the red arrows in Fig 9). Hence, the interpolating process of LMAR and NMAR leads to a smoothing effect of the corrected CT images and specific line profiles obtained in this study. Fig 11(D) and 11(E) also show that projection-based VMCT images can effectively eliminate cupping artifacts (red arrows) and bright/dark streaks (purple arrows). Moreover, the results of the projection-based VMI algorithm also yield the VMCT images and corresponding line profiles without the data smoothing processing. The experimental results of the line profiles indicated that the VMI-Atable algorithm performed better than the VMI-poly algorithm with respect to MAR in CT images. The object can be distinguished from the surrounding materials when the object is located in the position of the bright/dark streaks with the projection-based VMI algorithm (as indicated by the orange arrows in Fig 9).

To assess the image distortion affected by the metal artifacts, the circularity of the Cu rod in the CT images obtained using the different MAR methods was calculated. The ROI for the circularity calculation is shown in Fig 12(A). Fig 12(B)–12(F) show the CT images obtained using the various MAR methods. The Cu rod of the CT images obtained using the LMAR and NMAR methods revealed slight distortions (orange arrows), as shown in Fig 12(C) and 12(D), respectively. The image distortion was caused by the metal artifacts around the Cu rods. However, the projection-based VMCT images indicated that the shapes of the Cu rods were not

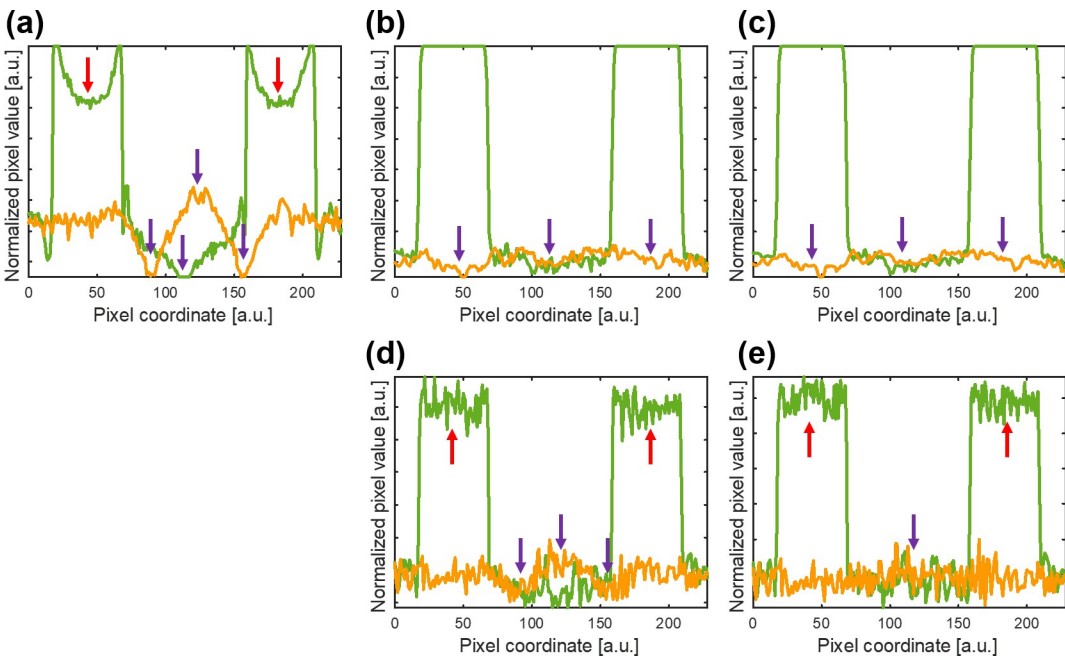

**Fig 11. Line profiles of the experimental CT images with the different MAR methods, as denoted in Fig 10.** Line profiles are represented by the green and orange lines of the corresponding uncorrected CT image. (a) The uncorrected CT image with 140 kV polychromatic X-ray. (b) The corrected CT images obtained using the LMAR method. (c) The corrected CT images obtained using the NMAR method. (d) The projection-based VMCT image obtained with a polynomial method (VMI-poly) at 100 keV. (e) The projection-based VMCT image obtained with the MLE and LUT method (VMI-Atable) at 100 keV.

affected by the metal artifacts. For quantitative analysis of the image distortion with the different MAR methods, the circularity of the Cu rod in the corresponding CT images was calculated, as presented in Table 4. The results revealed that the circularity of CT images obtained using the projection-based VMI algorithms (VMI-poly and VMI-Atable) was higher than those of the CT images obtained using the conventional MAR methods (LMAR and NMAR). In particular, the circularity of the CT image with the VMI-Atable algorithm was close to 1, which indicated that the Cu rod of the image was practically unaffected by the metal artifacts.

## Discussion and conclusions

In this study, a projection-based MD algorithm with linear MLE and LUT error correction (Atable) was implemented to generate VMCT images in order to reduce metal artifacts in the CT images. For comparison with the aforementioned algorithm, a conventional MD algorithm with polynomial approximation was also executed to create VMCT images. Furthermore, two conventional MAR methods with sinogram inpainting algorithms (LMAR and NMAR) were included in the comparison. The different MAR methods used for MAR were validated by PCD-CT simulation and experimental studies. In regards to the comparison with the conventional MAR methods, the simulated results of the CT images with XCAT phantom reveal that the projection-based VMCT images can reduce metal artifacts effectively without blurring the surrounding tissues at the position of the metal artifacts, which is particularly relevant for the case of the Atable-based algorithm used in the VMCT image. This is crucial when the observed object is located at the position of the metal artifacts. The analysis results of multiple quantitative indicators (PSNR, NRMSE, and SSIM) also demonstrate that the VMCT image with the Atable-based algorithm has better MAR performance than the other methods. The

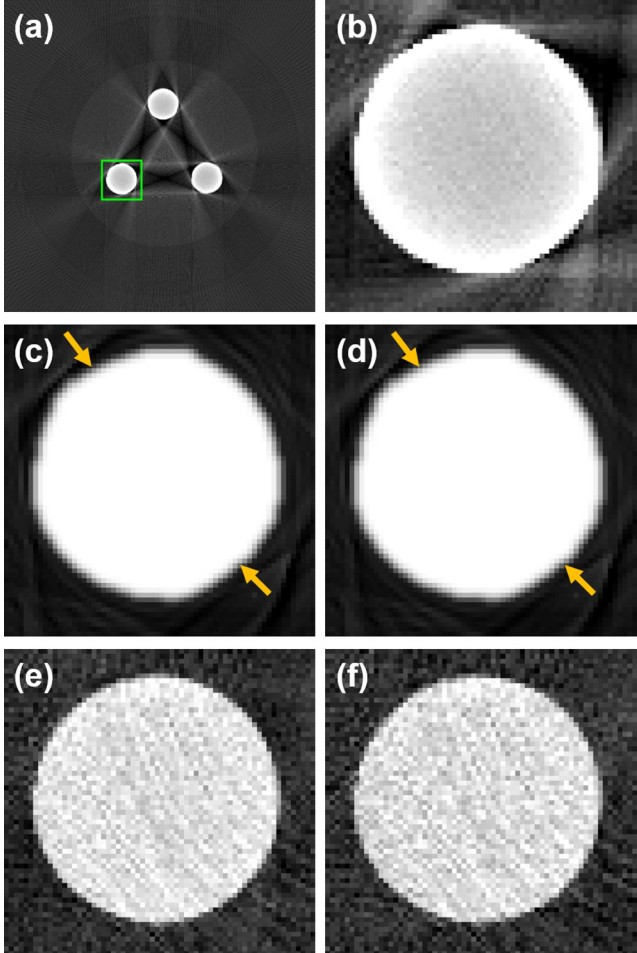

**Fig 12. Enlarged CT images from the experimental results using the MAE phantom with the different MAR methods.** (a) The ROI position for circularity calculations. (b) The uncorrected CT image with 140 kV polychromatic X-ray. (c) The corrected CT images obtained using the LMAR method. (d) The corrected CT images obtained using the NMAR method. (e) The projection-based VMCT image obtained using a polynomial method (VMI-poly) at 100 keV. (f) The projection-based VMCT image obtained using the MLE and LUT method (VMI-Atable) at 100 keV. Window width and level of the image display are 15,000 HU and 6,500 HU, respectively.

experimental results of the CT images and line profiles with the MAE phantom reveal that the different MAR methods considered in this study can effectively reduce metal artifacts. In regards to the line-profile results of the projection-based VMCT images, the Atable-based algorithm exhibits better MAR performance than the polynomial-based algorithm. Furthermore, image distortion around high-density materials can be avoided using projection-based VMCT images, which is critical in the measurement of the geometry of the metal implant in specific applications. In summary, the projection-based VMCT images obtained using the Atable algorithm can effectively reduce metal artifacts, while preventing object blurring at the metal artifact position and maintaining the metal implant geometry in the corrected CT

**Table 4. The circularity of the copper rod in the CT images obtained the different MAR methods.**

|  | LMAR | NMAR | VMI_poly | VMI_Atable |
|---|---|---|---|---|
| Circularity | 0.9697 | 0.9697 | 0.9800 | 0.9983 |

image. It may thus help improve the assessment of implants and bones, as well as soft tissue surrounding them, in postoperative follow-up examinations.

## Acknowledgments

The authors would like to thank Prof. Benjamin M. W. Tsui, Katsuyuki Taguchi, and Okkyun Lee for providing consultation on the PCD-CT imaging system setup and material decomposition algorithms.

## Author Contributions

**Conceptualization:** Chia-Hao Chang, Ching-Han Hsu, Hsin-Hon Lin.

**Data curation:** Hsiang-Ning Wu.

**Formal analysis:** Hsiang-Ning Wu.

**Investigation:** Chia-Hao Chang.

**Methodology:** Chia-Hao Chang, Hsin-Hon Lin.

**Software:** Hsiang-Ning Wu.

**Supervision:** Ching-Han Hsu, Hsin-Hon Lin.

**Validation:** Chia-Hao Chang.

**Visualization:** Chia-Hao Chang.

**Writing – original draft:** Chia-Hao Chang.

**Writing – review & editing:** Ching-Han Hsu, Hsin-Hon Lin.

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
