## [Decision Letter · Decision Letter 0]

14 Dec 2022

PONE-D-22-29123Virtual monochromatic imaging with projection-based material decomposition algorithm for metal artifacts reduction in photon-counting detector computed tomographyPLOS ONE

Dear Dr. Lin,

Thank you for submitting your manuscript to PLOS ONE. After careful consideration, we feel that it has merit but does not fully meet PLOS ONE’s publication criteria as it currently stands. Therefore, we invite you to submit a revised version of the manuscript that addresses the points raised during the review process.

We look forward to receiving your revised manuscript.

Kind regards,

Minsoo Chun, Ph.D.

Academic Editor

PLOS ONE

Journal Requirements:

"This work was supported in part by the Atomic Energy Council of Taiwan (No. AIE01030302). Hsin-Hon Lin was supported by National Science and Technology Council and Chang Gung University/Chang Gung Memorial Hospital (No. 111-2221-E-182 -008 -MY3 and CMRPD1K0442). The funders had no role in the study design, data collection, analysis, decision to publish, or preparation of the manuscript."

Reviewers' comments:

Reviewer's Responses to Questions

**Comments to the Author**

1. Is the manuscript technically sound, and do the data support the conclusions?

Reviewer #1: Partly

Reviewer #2: Yes

2. Has the statistical analysis been performed appropriately and rigorously? 

Reviewer #1: N/A

Reviewer #2: Yes

3. Have the authors made all data underlying the findings in their manuscript fully available?

Reviewer #1: Yes

Reviewer #2: Yes

4. Is the manuscript presented in an intelligible fashion and written in standard English?

Reviewer #1: No

Reviewer #2: Yes

5. Review Comments to the Author

Reviewer #1: We need to improve the English grammar of overall text in the paper.

The overall introduction and history of the research is impressive, almost like review paper. However, the importance of projection domain material decomposition is somewhat out of the ordinary.

Parameters of the developed CT

- The FOV of the developed machine seems a bit small.

- Why did you put PCD threshold at 30 to 80 KeV when it was 140kvp? The spectrum range is wider than that, and even with a filter, the photons in 30 under/80 upper spectrum range have inneglectable amount in a scope of beam hardening.

- There will be insufficient areas during the rebinning process with 360 degree bin. Based on the Nyquist frequency, considering the beam angle in the design, it is determined that the sampling rate of the periphery except the center is not sufficient. Considering that 1080 view was used in the numerical simulation, it is expected that the issue is clearly understood from the writer.

The windowing and image resolution of the Figure image are not sufficient to recognize the strike/beam-hardening artifact in the image. Additional work would be needed, such as narrowing windows width of the image.

In the linear summing/ray tracing method, how did you design the hardening/scatter that occurs as the beam is transmitted? Just simple integration of the path, or scatting effects were adopted?

Does each mass/linear attenuation coefficient values were referred from NIST database? or measured from the developed machine?

Through the calibration/test, what is the material range that can be reviewed during the calibration process? Is it possible to cover all the material ranges that can be used in the actual use-case?

We all know that models such as TASMIP/MASMIP/SPEKTR have limitations in mimicking the real world data.

It seems that unification or definition of the expression of the unit would be needed. (Kev, kv, kVp) it seems that a few typos of the unit were written.

How come did you used 100 KeV monochromatic energy as a reference of non-metal artifact image on 140 KVP polychromatic image? The average KeV in 140 KVP spectrum doesn’t seem as close to 100 KeV in TASMIP spectrum.

It seems that the full description of the abbreviation should be specified. (PSNR, NRMSE, etc) we all know that PSNR is Peak-Signal-To-Ratio, however, it may cause confusion to readers.

Reviewer #2: General Comments

Metal artifact reduction (MAR) is a major challenge in computed tomography (CT) to improve image quality and medical diagnosis and treatment. The aim of this study is to develop and implement two projection-based material decomposition (MD) algorithms to generate virtual monochromatic CT (VMCT) images for MAR, and evaluate their effectiveness in comparison with two conventional MAR methods, i.e., LMAR and NMAR. The derivation of the proposed methods, VMI-poly and VMI-Atable, are described in the manuscript. The materials and methods include a simulation study using the 3D XCAT phantom with titanimum (Ti) metal implants, and an experimental study using a spectral micro-CT prototype system equipped with a photon-counting detector (PCD) and a homemade metal artifact evaluation (MAE) phantom. Results from the simulation and experimental studies demonstrate that, in comparison to the conventional MAR methods, the projection-based VMCT images can not only reduce metal artifacts effectively but also simultaneously prevents object blurring at the metal artifact position and image distortion of the metal implants.

The manuscript addresses an important challenge in CT and is a timely study. It contributes greatly to the current research efforts in the development of MAR methods to improve CT image quality that will potentially lead to improvement in clinical diagnosis. In general, the different aspects of the study presented in manuscript are clearly described. The materials and methods used in the study are well designed and appropriate. The proposed projection-based MD algorithms and derivation of the VMCT images are clearly presented. Results from the simulation and experimental studies are well-presented and convincing. However, there are several specific comments and concerns listed below that need to be addressed. With satisfactory responses to the specific comments and concerns, the manuscript is recommended for publication in the PLOS ONE journal.

Specific Comments

1. Although in general both the scientific content and the writing of different sections of the manuscript are good, the organization of the manuscript can be improved. A typical scientific manuscript is usually arranged in sections in the order of introduction, materials and methods, results, discussion and conclusion. In this manuscript, some of the topics and sections are not placed in their logical order and can be re-arranged for easier reading by the readers.

2. Line 437. Please list the full letters of the acronyms, PSNR, NRME and SSIM.

3. Line 440. What do you mean by ‘gray values’?

4. Line 458. Please state what the ‘circle’ is in the sentence.

5. Line 466. Instead of ‘0 for highly non-circular shapes’, do you mean ‘larger deviation from 1 for greater non-circular shapes’?

6. Line 476. It would be good to replace ‘the conventional MAR methods and projection-based VMI algorithms for MAR’ by ‘the two conventional MAR methods, LMAR and NMAR, and the 2 projection-based VMI algorithms, VMI-Poly and VMI-Atable, for MAR’ at the beginning of the paragraph.

7. Line 485. It would be good to introduce the terms VMI-poly and VMI-Atable, in the earlier sections where they were first introduced and described.

8. Line 521. Should the ‘different ROIs’ be more clearly listed as ROI #1, ROI #2 and ROI #3 as shown in Figure 8.

9. Line 552 and Figs. 10 (b) – (e). The gray-scale of the image are too saturated to see any variations within the images of the different rods.

10. Lines 672 – 674. The sentence does seem to make sense. Please rewrite it.

6. PLOS authors have the option to publish the peer review history of their article (what does this mean?). If published, this will include your full peer review and any attached files.

Reviewer #1: No

Reviewer #2: No

---

## [Decision Letter · Decision Letter 1]

27 Feb 2023

Virtual monochromatic imaging with projection-based material decomposition algorithm for metal artifacts reduction in photon-counting detector computed tomography

PONE-D-22-29123R1

Dear Dr. Lin,

We’re pleased to inform you that your manuscript has been judged scientifically suitable for publication and will be formally accepted for publication once it meets all outstanding technical requirements.

Kind regards,

Minsoo Chun, Ph.D.

Academic Editor

PLOS ONE

Additional Editor Comments (optional):

Reviewers' comments:

Reviewer's Responses to Questions

**Comments to the Author**

1. If the authors have adequately addressed your comments raised in a previous round of review and you feel that this manuscript is now acceptable for publication, you may indicate that here to bypass the “Comments to the Author” section, enter your conflict of interest statement in the “Confidential to Editor” section, and submit your "Accept" recommendation.

Reviewer #1: All comments have been addressed

Reviewer #2: All comments have been addressed

2. Is the manuscript technically sound, and do the data support the conclusions?

Reviewer #1: Yes

Reviewer #2: Yes

3. Has the statistical analysis been performed appropriately and rigorously? 

Reviewer #1: Yes

Reviewer #2: Yes

4. Have the authors made all data underlying the findings in their manuscript fully available?

Reviewer #1: Yes

Reviewer #2: Yes

5. Is the manuscript presented in an intelligible fashion and written in standard English?

Reviewer #1: Yes

Reviewer #2: Yes

6. Review Comments to the Author

Reviewer #1: Thank you for your sincere response to my overall review comment. I don't agree with all the answers, but as a result, I think the theory and grounds of the paper you submitted have become sound and stronger. The thesis at this point is considered acceptable enough.

Reviewer #2: The authors have responded to this reviewer's concerns and comments satisfactorily. The manuscript is recommended for publication in PLOS ONE.

7. PLOS authors have the option to publish the peer review history of their article (what does this mean?). If published, this will include your full peer review and any attached files.

Reviewer #1: No

Reviewer #2: No

---

## [Editor Report · Acceptance letter]

3 Mar 2023

PONE-D-22-29123R1 

Virtual monochromatic imaging with projection-based material decomposition algorithm for metal artifacts reduction in photon-counting detector computed tomography 

Dear Dr. Lin:

I'm pleased to inform you that your manuscript has been deemed suitable for publication in PLOS ONE. Congratulations! Your manuscript is now with our production department. 

Kind regards, 

on behalf of

Dr. Minsoo Chun 

Academic Editor

PLOS ONE